# Modeling spatial determinants of initiation of breastfeeding in Ethiopia: A geographically weighted regression analysis

**Samuel Hailegebreal**[1]\*, **Yosef Haile**[2], **Binyam Tariku Seboka**[3], **Ermias Bekele Enyew**[4], **Tamiru Shibiru**[5], **Zeleke Abebaw Mekonnen**[6], **Shegaw Anagaw Mengiste**[7]

**1** Department of Health Informatics, College of Medicine and Health Science, School of Public Health, Arba Minch University, Arba Minch, Ethiopia, **2** School of Public Health, College of Medicine and Health Sciences, Arba Minch University, Arba Minch, Ethiopia, **3** Department of Health Informatics, College of Medicine and Health Sciences, Dilla University, Dilla, Ethiopia, **4** Department of Health Informatics, Institute of Public Health, Mettu University, Mettu, Ethiopia, **5** School of Medicine, College of Medicine and Health Sciences, Arba Minch University, Arba Minch, Ethiopia, **6** Health System Directorate, Ministry of Health, Addis Ababa, Ethiopia, **7** University of South-Eastern Norway, Kongsberg, Norway

\* samuastd@gmail.com

## Abstract

### Background

The World Health Organization (WHO) encourages breastfeeding to begin within the first hour after birth in order to save children's lives. In Ethiopia, different studies are done on the prevalence and determinants of breastfeeding initiation, up to our knowledge, the spatial distribution and the spatial determinants of breast feeding initiation over time are not investigated. Therefore, the objectives of this study were to assess spatial variation and its spatial determinant of delayed initiation of breastfeeding in Ethiopia using Geographically Weighted Regression (GWR).

### Methods

A cross-sectional study was undertaken using the nationally representative 2016 Ethiopian Demographic and Health Survey (EDHS) dataset. Global Moran's I statistic was used to measure whether delayed breastfeeding initiation was dispersed, clustered, or randomly distributed in study area. Ordinary Least Squares (OLS) regression was used to identify factors explaining the geographic variation in delayed breastfeeding initiation. Besides, spatial variability of relationships between dependent and selected predictors was investigated using geographically weighted regression.

### Result

A total weighted sample of 4169 children of aged 0 to 23 months was included in this study. Delayed initiation of breastfeeding was spatially varies across the country with a global Moran's I value of 0.158 at (p-value<0.01). The hotspot (high risk) areas were identified in the Amhara, Afar, and Tigray regions. Orthodox religion, poor wealth index, caesarian section,

**Data Availability Statement:** The data we used for this study is available in the DHS program. A letter of approval for the use of the data was secured

from the Measure DHS program and the data set was downloaded from the website www. measuredhs.com (https://dhsprogram.com/data/available-datasets.cfm). We used EDHS 2016 Kids data set (KR file) and extracted the outcome variable (Breast feeding initiation) and explanatory variables. The location data (latitude and longitude coordinates) was also taken from selected enumeration areas (clusters).

**Funding:** The authors received no specific funding for this work.

**Competing interests:** The authors declare that they have no competing interests.

**Abbreviations:** CI, Confidence Interval; CS, Caesarian Section; CSA, Central Statistical Agency; EDHS, Ethiopia Demographic and Health Survey; LLR, Log-Likelihood Ratio; OLS, Ordinal Least Squares; SNNPR, Southern Nations, Nationalities, and Peoples' Region.

baby postnatal checkup, and small size of a child at birth were spatially significant factors for delayed breastfeeding initiation in Ethiopia.

## Conclusion

In Ethiopia initiation of breastfeeding varies geographically across region. A significant hot-spot was identified in the Amhara, Afar, and Tigray regions. The GWR analysis revealed that orthodox religion, poor wealth index, caesarian section, baby postnatal checkup, and small birth weight were spatially significant factors.

## Introduction

Despite the fact that World Health Organization (WHO) and United Nations International Children's Emergency Fund (UNICEF) recommend starting breastfeeding within one hour of birth, many babies are not breastfed in the first hour of life [1]. Globally, 78 million babies, or three out of every five, are not breastfed within the first hour of life, leading to an increased risk of death and suffering [2, 3]. Early initiation of breastfeeding (EIBF) is an important pathway for reducing malnutrition and preventing mortality for young children [4, 5] and reducing the risk of postpartum hemorrhage for the mothers [6].

Previous studies has shown that newborns who began breastfeeding after the first hour of birth had a 33% higher risk of dying than those who began breastfeeding within the first hour of birth [7]. In low- and middle-income countries (LMICs), the overall prevalence of delayed breastfeeding initiation is 53.8%, ranging from 15.0% in Burundi to 83.4% in Guinea [8]. According to studies conducted in Uganda and Bangladesh, nearly half (48%) and about three-fifths of mothers initiated breastfeeding later than one hour after birth, respectively [9–11]. Findings from the northern part of Ethiopia revealed that 21.2% of newborns delayed breastfeeding initiation [12].

Several studies suggest that factors such as religion, wealth index, cesarean delivery, Antenatal care (ANC), maternal complications during pregnancy, a lack of postnatal/neonatal care guidelines at hospitals, home delivery, birth weight, birth order, parity, employment status, child sex, and place of residence parental education are associated with a delay in breast feeding initiation [9–18]. However, breastfeeding initiation have been found to vary across geographical locations [19, 20].

Therefore, this study aimed to assess the regional variation and model the spatial determinants of delayed breastfeeding initiation in Ethiopia. The findings of this study may provide insight for authorities, researchers, and health professionals on the country's delayed initiation of breastfeeding situation, allowing for targeted interventions in areas where delayed initiation of breastfeeding is prevalent.

## Methods

### Study design, setting and population

A cross-sectional study was undertaken using the nationally representative 2016 Ethiopian Demographic and Health Survey (EDHS) dataset. Ethiopia is located (3˚-14˚N, 33˚ – 48˚E) in the Horn of Africa. It has 9 regional states (Afar, Amhara, Benishangul-Gumuz, Gambela, Harari, Oromia, Somali, Southern Nations, Nationalities, and People Region (SNNPR), and Tigray regions) and two cities (Addis Ababa and Dire-Dawa) administrations every five years (Fig 1). We used the 2016 EDHS data for this study which was conducted from January 18 2016 to

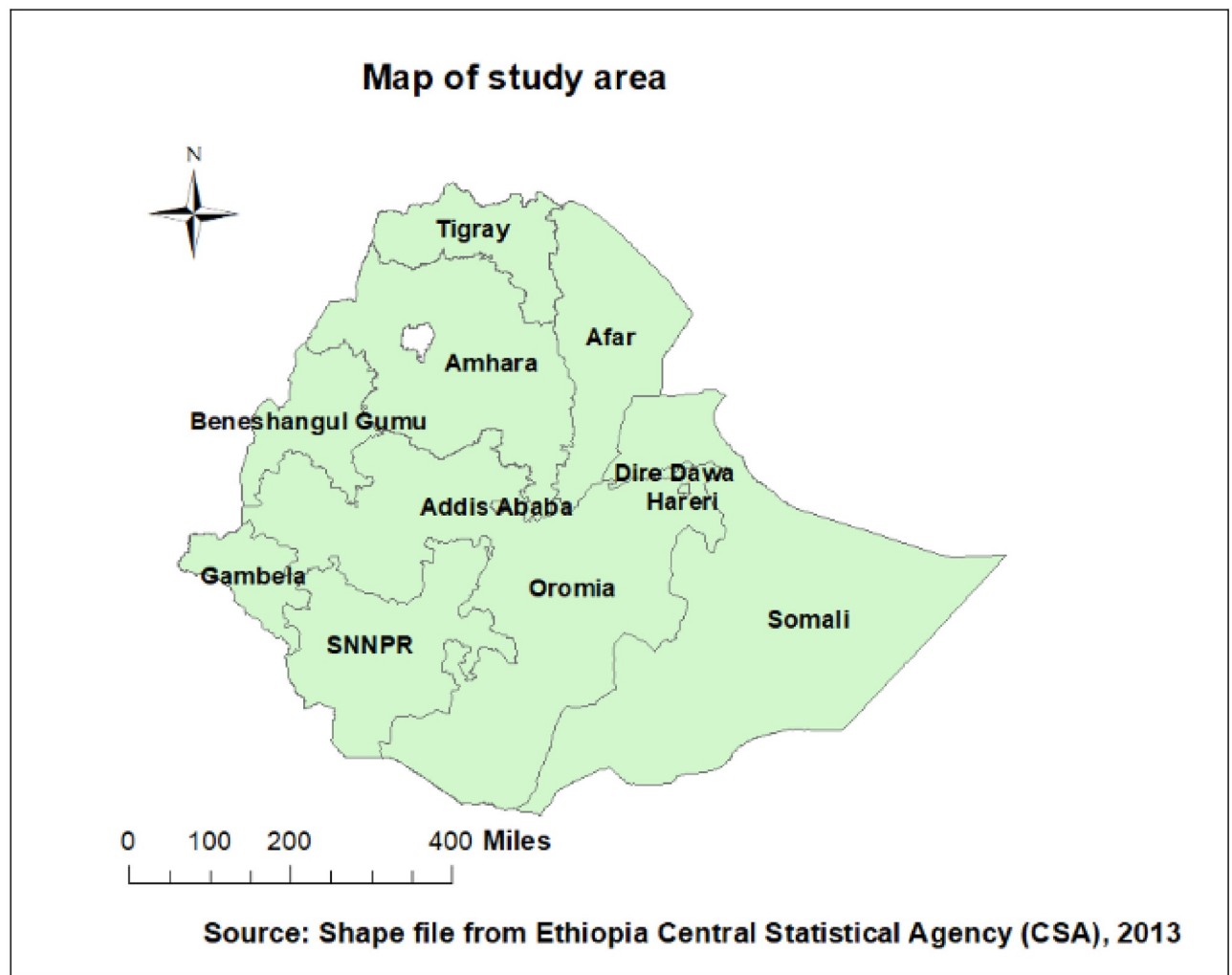

**Fig 1. Map of study area, Ethiopia.**

June 27, 2016. In EDHS 2016, a two-stage stratified cluster sampling technique was employed using the 2007 Population and Housing Census (PHC) as a sampling frame. In the first stage, 645 EAs (202 in the urban area) were selected, and in the second stage, on average 28 households were systematically selected. We got the data from the EDHS dataset, which is only available (www.dhsprogram.com) through site requests. For this study, kids data set was used with a total weighted sample of 4169 women who ever breastfeed and who had living children less than 2 years of age. The full EDHS report included the detailed sampling technique [21].

## Study variables

The outcome variable was delayed initiation of breastfeeding. It is put to the breast within the first hour of birth. It was measured based self-report of the mother and classified as "early" if it began within one hour, and "late /delayed" if it began later than one hour.

## Independent variables

The variables were selected based on previous literatures review [22–26]. In this analysis variables were recoded as follow mother's age ("15–24 years","25–34 years","35–49 years"), marital

status ("married", "unmarried"), parity ("Primiparous", "multiparous", "grand multiparous"), place of residence ("rural", "urban"), educational status ("no education", "primary", "secondary or above"), working status ("not working", "working"), religion ("orthodox", "Muslim", "protestant", "others"), household wealth ("poor", "middle", "richer"), child age ("0–5 months", "6–11 months", "12–23 months"), baby post-natal check ("yes"/ "no"), antenatal care ("yes"/"no"), place of birth ("home", "health facility"), mode of delivery ("caesarean section", "vaginal"), birth weight ("small", "average" "large"), birth order ("1–3","4–6" and "above 6"), media exposure ("yes"/ "no") and child sex ("male" "female").

## Data management and statistical analysis

STATA version 14 statistical software was used to execute descriptive analysis. The spatial analysis was carried out with ArcGIS 10.7. The weighted proportions of outcome variable and potential predictor variables were tabulated in STATA and exported to excel before being transferred to ArcGIS 10.7 for further analysis. When a variable has a "missing value," it should have a response but does not, either because the interviewer accidentally left out the question or because the respondent declined to answer. We remove missing values from our analysis by using the STATA drop command in combination with a logical or conditional statement.

## Spatial analysis

ArcGIS V.10.7 software was used for the spatial analysis to determine whether the pattern was clustered, dispersed, or random across the study area [27], and SaTScan V.9.6 software was used for the local cluster analysis. Global Moran's I is a spatial statistic that measures spatial autocorrelation by taking the entire data set and generating a single output value ranging from -1 to +1. Moran's, I value close to -1 indicates that delayed breastfeeding initiation is dispersed, whereas Moran's I close to +1 indicates that delayed breastfeeding initiation is clustered, and Moran's I close to 0 indicates that delayed breastfeeding initiation is randomly distributed. Moran's, I value that were statistically significant ($p < 0.05$) had a chance to reject the null hypothesis, indicating the presence of spatial autocorrelation. Using Getis-Ord Gi* statistics, the local spatial analysis was performed to identify specific significant hot spot and cold spot areas.

## Spatial regression

### Ordinary Least Squares (OLS)

Spatial regression modeling was used to identify predictors of the spatial variation of delayed breastfeeding initiation in study area. OLS is a global statistical model that is used to test and explain the relationship between the dependent and independent variables [28]. The OLS was used as a diagnostic tool as well as to select the appropriate predictors (in terms of their relationship with delayed breastfeeding initiation) for the Geographic Weighted Regression (GWR) model [29]. The Koenker Bp technique was used to see if the model could be used to do a spatially weighted regression analysis. When the Koenker statistics are significant (p-value<0.05), the GWR analysis is examined, which suggests the relationships between the dependent and independent variables change from place to place. The coefficients of explanatory variables in a correctly constructed OLS model should be statistically significant and have either a positive or negative sign. Multicollinearity (VIF<10) was also tested to rule out redundancy among independent variables.

## Geographically weighted regression

GWR is a spatial regression technique that uses a regression equation to fit to each feature in a spatial dataset to provide a local model for understanding/predicting from a set of independent variables [30]. So, after that, we used exploratory regression with the appropriate tests to justify the assumptions. The GWR model [31] can be expressed as follows:

$$Yi = \boldsymbol{\beta}_{0(u_i,\ v_i)} + \sum\nolimits_{k=1}^{p} \boldsymbol{\beta}_k(\boldsymbol{u_i v_i})\, \boldsymbol{X_{ik}} + \in_i$$

where $Y_i$ are observations of response $Y$, $u_i v_i$ are geographical points (longitude, latitude), $\beta_K$ $(u_i v_i)$ (k = 0, 1 . . . p) are p unknown functions of geographic locations, $u_i v_i$, $X_{ik}$ are explanatory variables at location, $u_i v_i$, $i = 1, 2, . . .$ n and $\in_i$ are error terms/residuals with zero mean and homogenous variance ($\sigma 2$). The GWR equation is calibrated using data from nearby features, whereas the OLS equation uses data from all features. Finally, the corrected Akaike Information Criteria (AICc) and adjusted R-squared were used to compare models. The model with the lowest AICc value and the highest adjusted R-squared value was determined to be the best fit for the data.

## Result

### Characteristics of the respondents and study children

A total weighted sample of 4169 children of aged 0 to 23 months was included in this study. More than half (52%) of the children were females. The majority (58.46%) of the mothers were in the age group of 15–29 years. Most, 1841(44.16%) and 852 (20.45%) of study participants were from Oromia and Southern Nations, Nationalities and Peoples' Region (SNNPR) respectively. About 1899 (45.55%) and 1409 (33.80%) mothers belong to the poor and rich household index quintiles, respectively (Table 1).

### Prevalence of delayed initiation of breastfeeding in Ethiopian

In the current study, the overall prevalence of delayed breastfeeding was 24.22% [95% CI: 22.94%, 25.55%]. The highest percentage of delayed breastfeeding was 56% [95% CI: 46%, 66%] seen in the Afar region (Fig 2).

### Spatial autocorrelation

The spatial distribution of delayed initiation of breastfeeding among children aged 0–23 months showed significant spatial variation across the country with a global Moran's I value of 0.158 (p-value<0.01) (Fig 3).

### Hot spot (Getis-Ord Gi*) analysis

The statistically significant hotspot (high risk) areas of delayed initiation of breastfeeding were identified in the Amhara, Afar, and Tigray regions. While significant cold spot (low risk) areas were detected in the Eastern SNNPRs, southern and eastern Oromia, Dire Dawa, Harari regions (Fig 4).

### Spatial scan statistics

A total of 276 significant clusters were identified using spatial scan analysis. The most likely (primary) clusters were located in Afar, Tigray, Amhara, central Oromia, Addis Ababa, and Benishangul-Gumuz at (14.222399 N, 38.163618 E) / 591.55 km radius. Children aged 0–23 months who lived in the primary cluster were 2.2 times more likely than those who lived

**Table 1. Socio-demographic characteristics of respondents and newborns.**

| Variables | Frequency | Percent (%) |
|---|---|---|
| **Mother age** | | |
| 15–29 | 2437 | 58.46 |
| 30–39 | 1474 | 35.36 |
| 40–49 | 258 | 6.18 |
| **Religion** | | |
| Orthodox | 1421 | 34.09 |
| Muslim | 1733 | 41.58 |
| protestant | 866 | 20.77 |
| others | 148 | 3.56 |
| **Child sex** | | |
| Male | 2,010 | 48.22 |
| Female | 2,159 | 51.78 |
| **Child age in month** | | |
| 0–5 | 1182 | 28.36 |
| 6–11 | 1070 | 25.67 |
| 12–23 | 1916 | 45.96 |
| **Educational level** | | |
| No education | 2515 | 60.32 |
| Primary education | 1284 | 30.81 |
| Secondary and above | 370 | 8.88 |
| **Wealth index** | | |
| Poor | 1899 | 45.55 |
| Middle | 861 | 20.65 |
| Rich | 1409 | 33.80 |
| **Marital status** | | |
| Single | 138 | 3.31 |
| Married | 4031 | 96.69 |
| **Place of residence** | | |
| Urban | 495 | 11.87 |
| Rural | 3674 | 88.13 |
| **Region** | | |
| Tigray | 307 | 7.36 |
| Afar | 41 | 0.98 |
| Amhara | 769 | 18.44 |
| Oromia | 1841 | 44.16 |
| Somali | 170 | 4.08 |
| Benishangul | 44 | 1.06 |
| SNNPR | 852 | 20.45 |
| Gambela | 10 | 0.23 |
| Harari | 10 | 0.24 |
| Addis Ababa | 107 | 2.57 |
| Dire dawa | 18 | 0.42 |

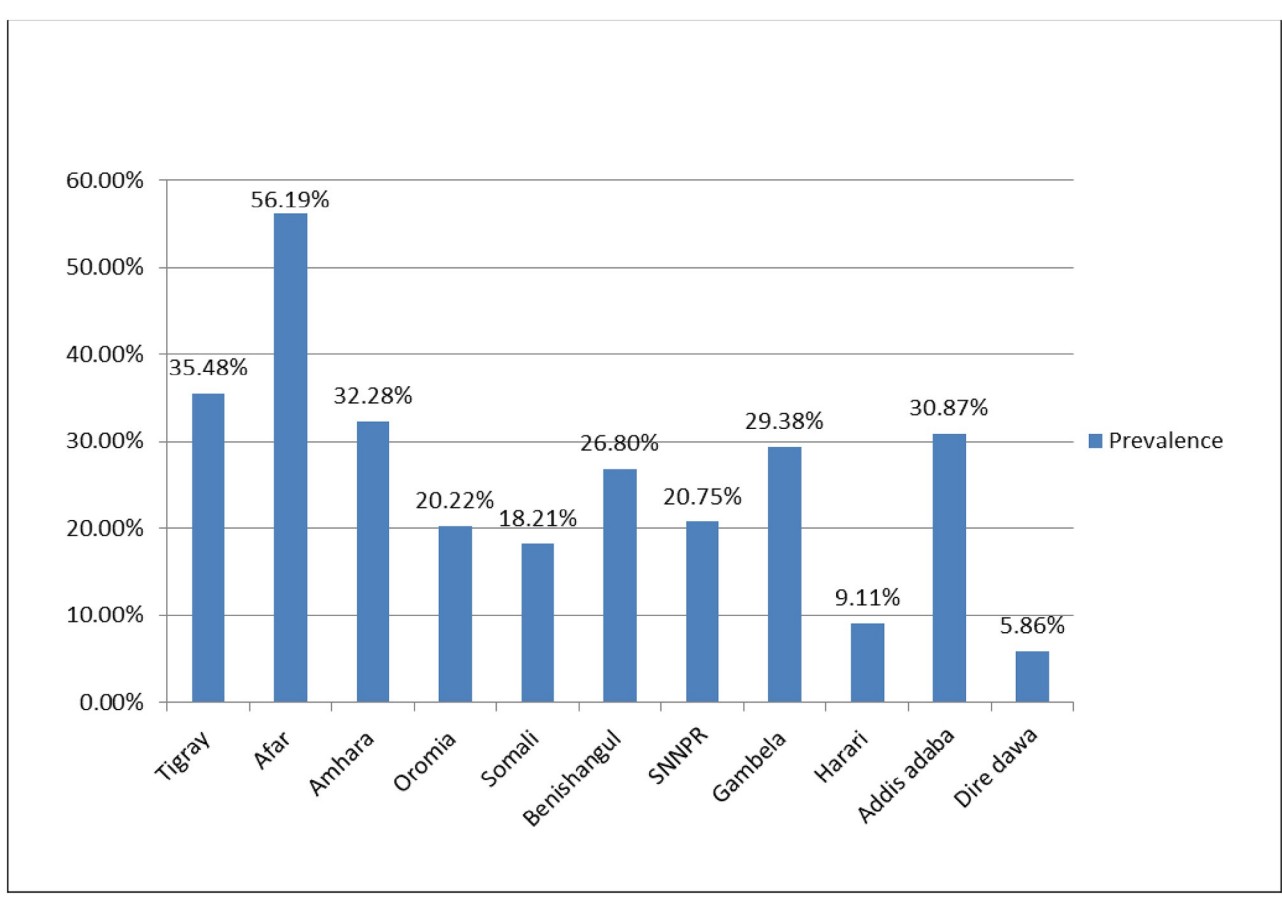

**Fig 2. Prevalence of delayed initiation of breastfeeding across regions in Ethiopia, 2016.**

outside the window to experience delayed breastfeeding initiation (RR = 2.19, LLR = 107.39, P-value 0.001) (Fig 5).

### Factors affecting the spatial variation of delayed breastfeeding

**Ordinary least square regression.** For the candidate explanatory variables, ordinal least squares (OLS) model was fitted. All of the OLS requirements were met in this model. The OLS model was validated to detect multicollinearity among the independent predictors, with a mean VIF of less than 10. The Joint F-statistics and Joint Wald statistic were statistically significant (p<0.01), shows that the model was significant. The model explained 14% of the variation in delayed breastfeeding, rendering to the adjusted R2. The Koenker statistics were statistically significant (p<0.01), indicating a non-stationary between the independent variables and the dependent variable across the study areas. This suggests that GWR should be used because it considers that the relationship between independent and dependent variables is spatially heterogeneous across area (Table 2).

**Geographically weighted regression.** The global (OLS) regression model revealed that determinants of delayed breastfeeding initiation hot areas. Moreover, OLS implies that the relation between each explanatory variable and the dependent is constant/stationary across the study area; we used GWR to improve the model in cases where the predictors were not stationary. Since, the GWR model has a higher adjusted R2 and a lower Akaike's Information

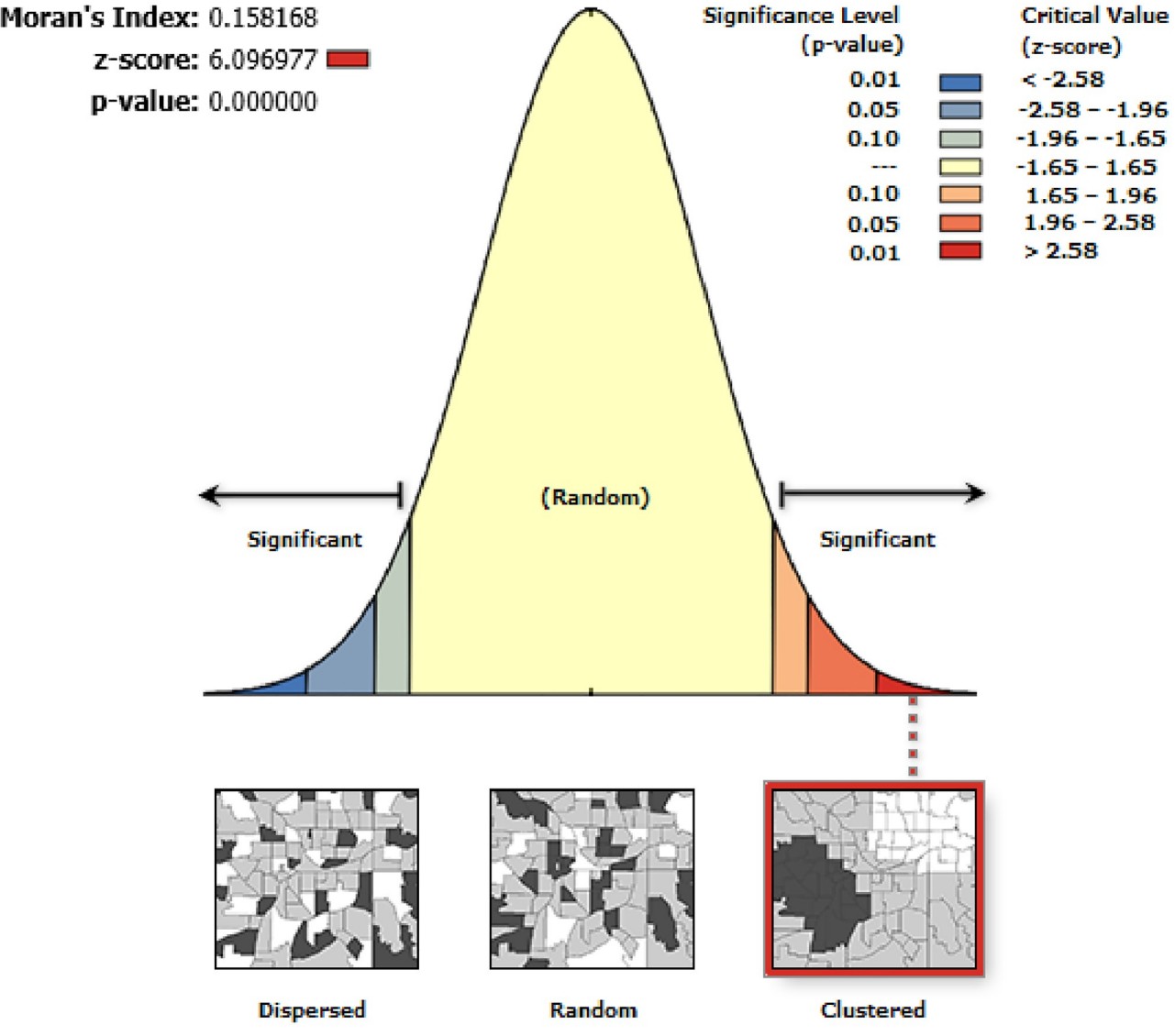

**Fig 3. The global spatial autocorrelation analysis delayed initiation of breastfeeding in Ethiopia.**

Criterion (AIC) value than the OLS model, its ability to explain delayed breastfeeding initiation has improved (Table 3). The strength of the relationship with independent variables varies spatially, and variable effects have both positive and negative spatial impacts.

In the GWR mode, being orthodox religion, poor wealth index, caesarian section, baby postnatal checkup, and small birth weight were spatially significant factors for delayed breastfeeding initiation in Ethiopia. Being orthodox religion follower had a positive and negative relationship with delayed initiation of breastfeeding with the coefficient ranging from -0.585 to 0.225, implies the effect of association varies across region. As shown in Fig 6, orthodox religion had strong positive predictor of delayed breastfeeding in the Harari, Dire Dawa, Eastern Oromia and Somali Region. On the other hand, the negative and strong relationship between orthodox religion and delayed initiation of breastfeeding was observed in Tigray, Gambela, and the northern part of Amhara (Fig 6).

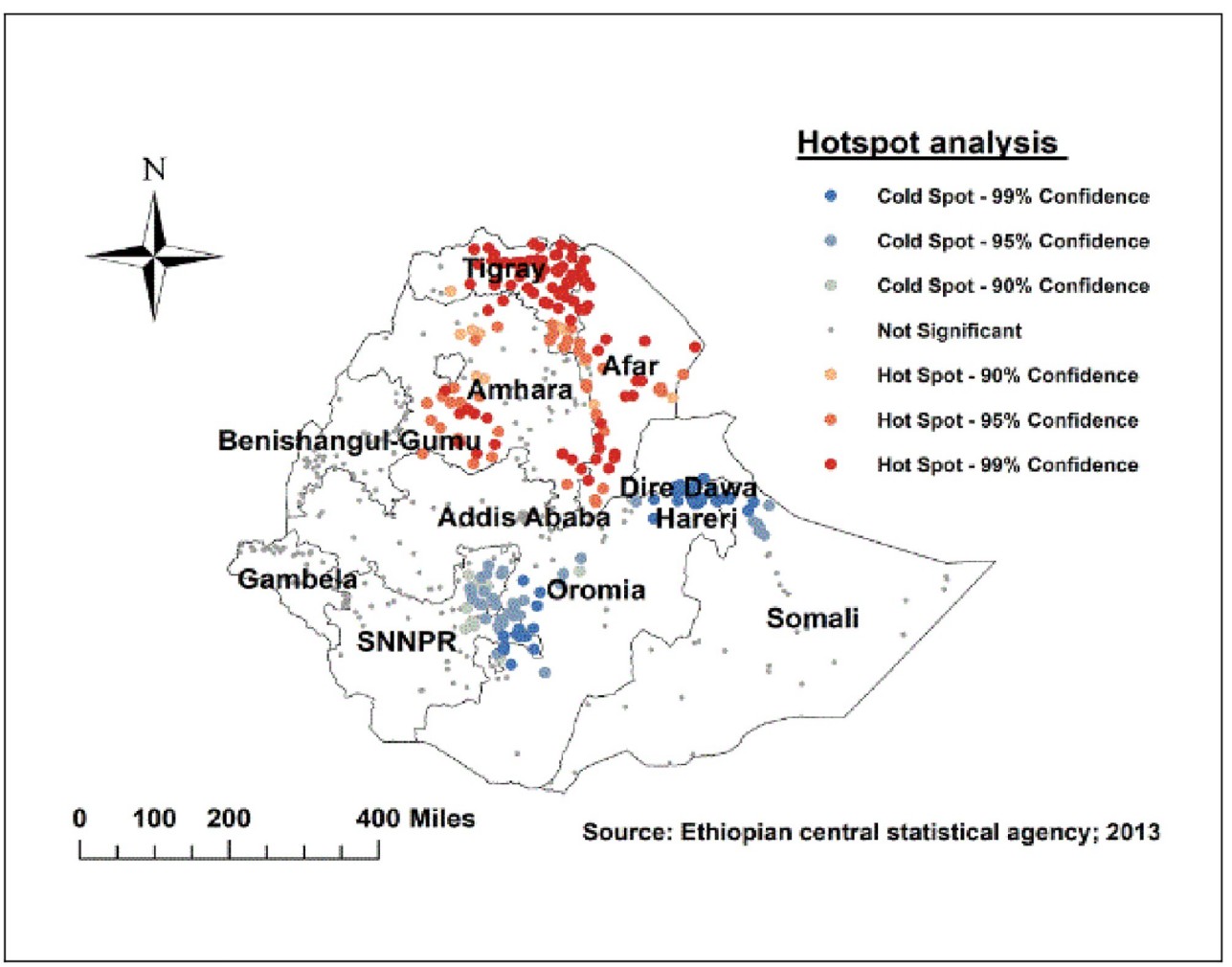

**Fig 4. Hotspot analysis of delayed initiation of breastfeeding in Ethiopia, 2016.**

This finding also highlights the spatial variation in relationship between delayed initiation of breastfeeding and wealth index. Being mother from poor wealth status showed a strong and positive relationship with delayed initiation of breastfeeding in the Tigray, Amhara Afar, Southern Somalia, Oromia and Addis Ababa (Fig 7). Moreover, mothers who delivered by caesarian section had a strong and positive relationship with delayed initiation in Tigray, and border of Afar regions (Fig 8).

As shown in Fig 9 baby receiving a postnatal check also a significant predictor of delayed initiation of breastfeeding. The strong and positive relationship was found in Oromia, SNNPR, Addis Ababa, Southeastern part of Amhara, southwestern part of Afar, and southwestern Somali region (Fig 9).

Furthermore, being small birth weight at birth was spatial predictor of delayed initiation of breastfeeding in eastern Somali region, southern and eastern parts of Afar, Dire Dawa, and Harari (Fig 10).

## Discussion

This study aimed to explore the spatial clustering and spatial determinants of delayed initiation of breastfeeding in Ethiopia. This study revealed, the overall prevalence of delayed

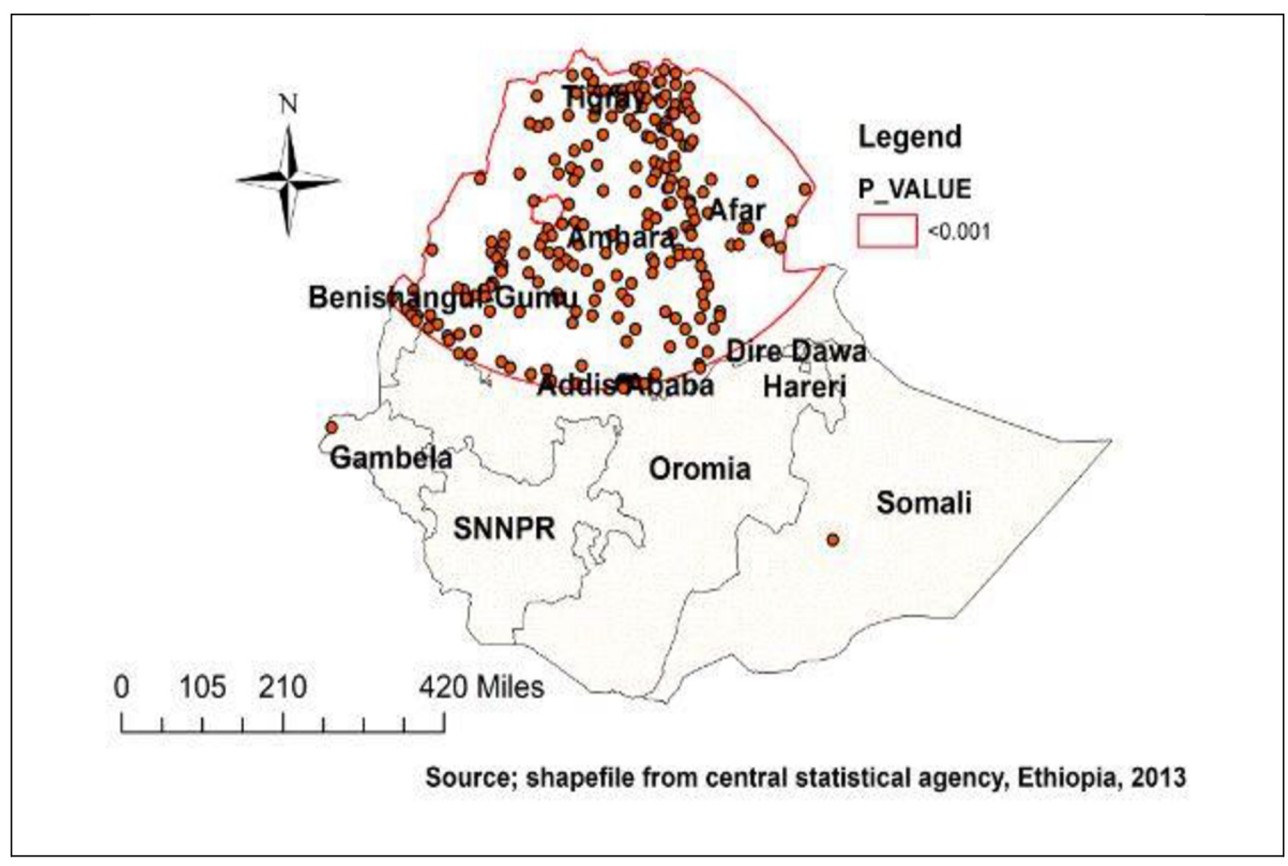

**Fig 5. Most likely (primary) cluster for delayed initiation of breastfeeding in Ethiopia.**

breastfeeding was 24.22% [95% CI: 22.94%, 25.55%]. This finding was lower than a study conducted in Uganda [10], Bangladesh [32], Ethiopia [23], and South Sudan [14]. These disparities could be explained by differences in study participants' health-care utilization, culture, and socioeconomic status. Across region highest and the lowest proportion of delayed

**Table 2. Summary of ordinary least squares result.**

| Variable | Coefficient | Robust standard error | Robust t statistics | Robust probability | VIF |
|---|---|---|---|---|---|
| Intercept | 0.06 | 0.020 | 3.09 | < 0.01 | ------ |
| Orthodox | 0.12 | 0.024 | 5.05 | < 0.01 | 1.13 |
| Proportion poor | 0.17 | 0.028 | 6.09 | < 0.01 | 1.27 |
| Proportion of cesarean delivery | 0.22 | 0.079 | 2.83 | < 0.01 | 1.12 |
| Proportion of baby postnatal checkup | 0.13 | 0.063 | 2.08 | < 0.05 | 1.09 |
| Proportion of small size at birth | 0.20 | 0.046 | 4.24 | < 0.01 | 1.08 |
| **Ordinary least square regression diagnostics** | | | | | |
| Number of observation | 611 | Adjusted R-Squared | | | 0.137 |
| Joint F-Statistic | 20.37 | Prob(>F), (5,605) degrees of freedom | | | < 0.01 |
| Joint Wald Statistic: | 100.27 | Prob(>chi-squared), (5) degrees of freedom | | | < 0.01 |
| Koenker (BP) Statistic | 22.100 | Prob(>chi-squared), (5) degrees of freedom | | | < 0.01 |
| Jarque-Bera Statistic | 39.838 | Prob(>chi-squared), (2) degrees of freedom | | | < 0.01 |

**Table 3. Model comparison of OLS and GWR model.**

| Model comparison | OLS | GWR |
|---|---|---|
| Akaike's Information Criterion (AICc) | -45.49 | -89.36 |
| Adjusted R-square | 0.14 | 0.23 |

breastfeeding was seen in the Afar and Dire Dawa region respectively. The findings were similar to the spatial analysis conducted in this study and previous study [33]. The possible reason might be mothers from the metropolitan area may have a higher level of education and have better access to breastfeeding knowledge. Besides, residents of metropolitan cities are completely urbanized. This makes media, health services, and health education more accessible. The improved infrastructure in metropolitan areas has a positive impact on access to health services [34].

Delayed initiation of breastfeeding spatially varies across the country with a global Moran's I value of 0.158 at (p-value<0.01). The hotspot areas were identified in the Amhara, Afar, and Tigray regions. Whereas, cold spot (low risk) areas were detected in the Eastern SNNPR, southern and eastern Oromia, Dire Dawa, Harari regions. This could be attributed to the fact that cultural variation, campaigns promoting baby formula, variation in health service

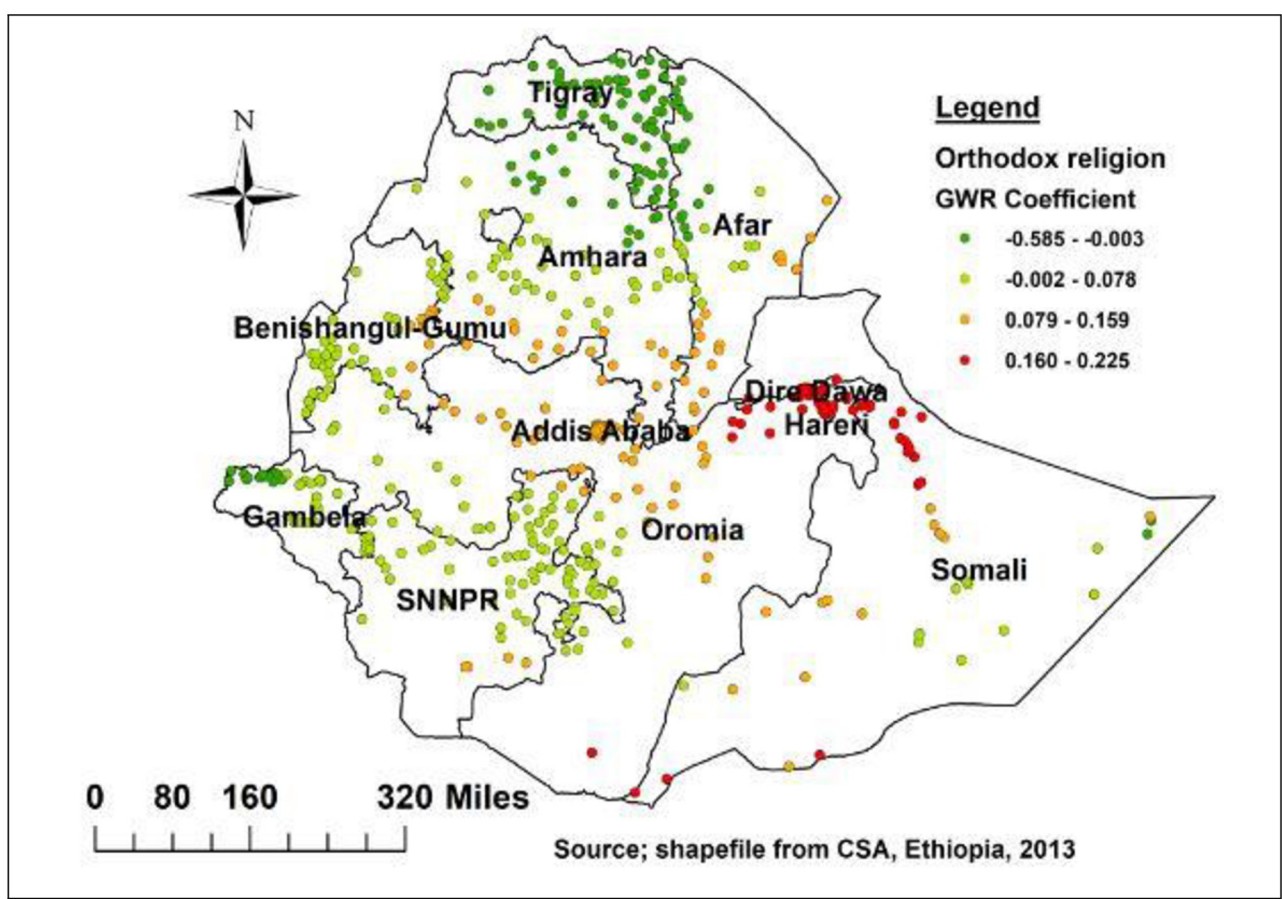

**Fig 6. Geographically weighted regression coefficients for orthodox religion to predict the hotspots of delayed breastfeeding in Ethiopia.**

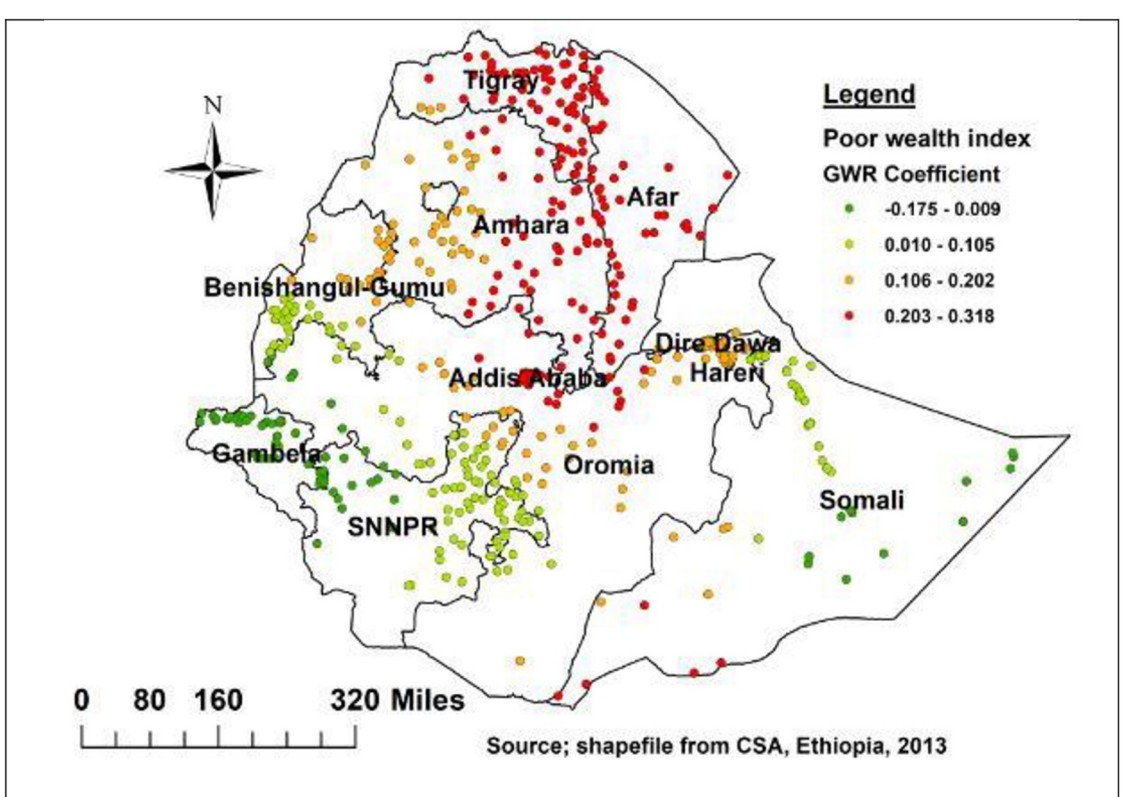

**Fig 7. Poor wealth index geographic weighted regression coefficients to predict delayed breastfeeding in Ethiopia.**

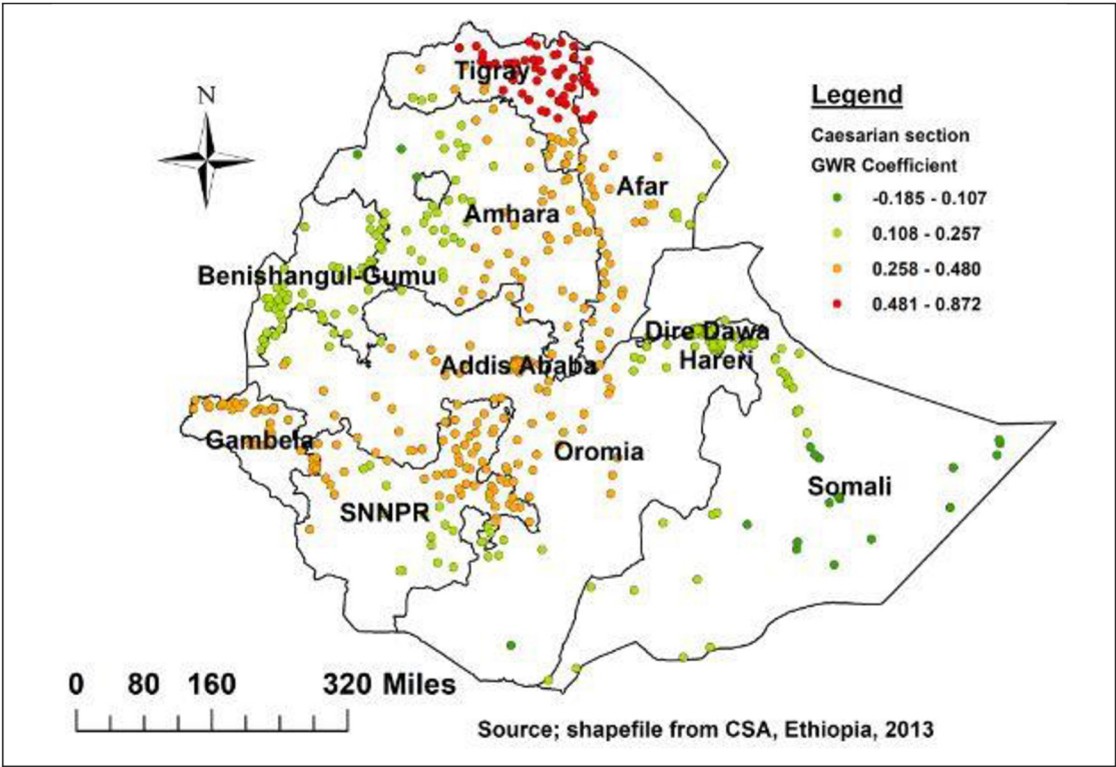

**Fig 8. Caesarian section geographic weighted regression coefficients to predict delayed breastfeeding in Ethiopia.**

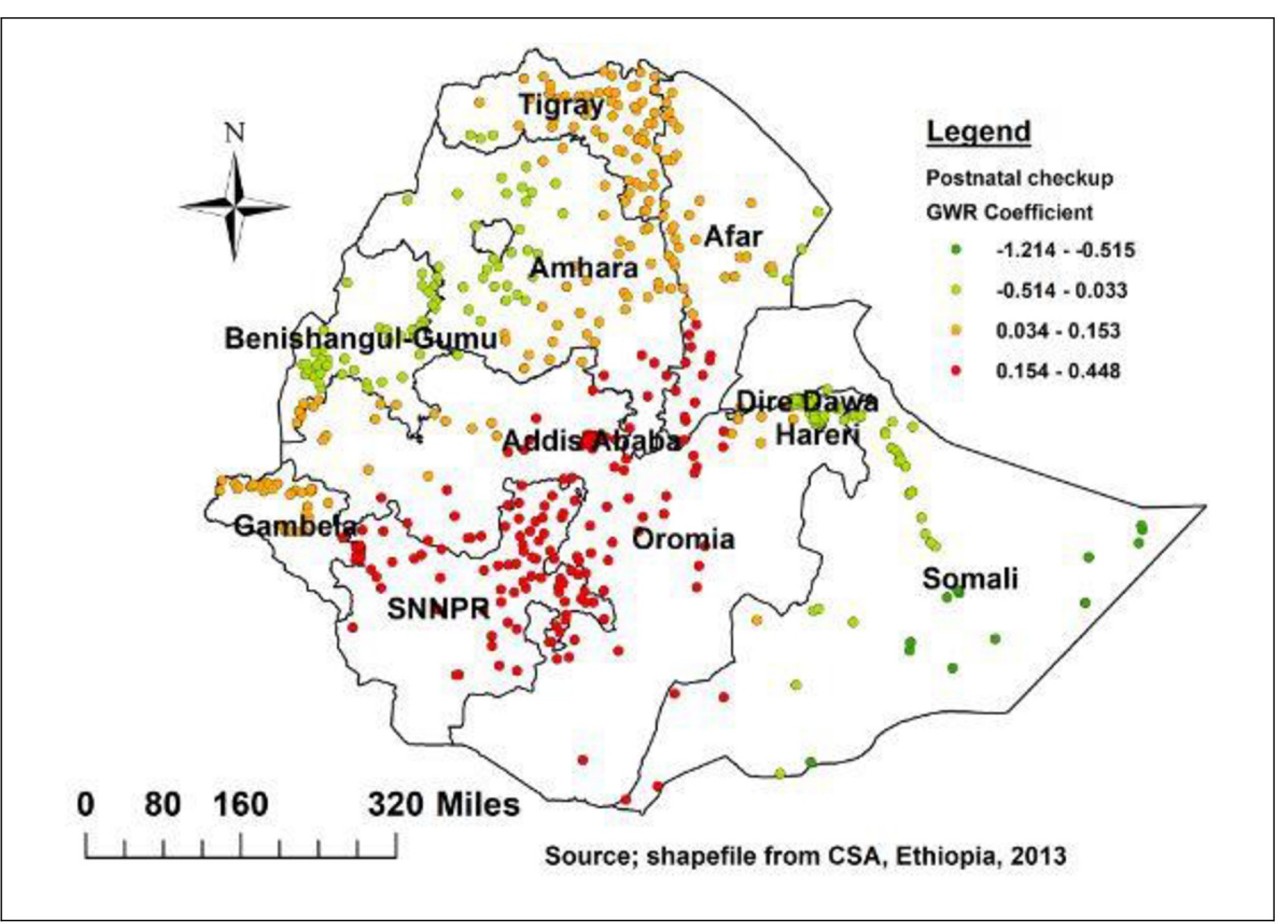

**Fig 9. Baby postnatal checkup geographic weighted regression coefficients to predict delayed breastfeeding in Ethiopia.**

utilization across the regions [24, 33, 35, 36]. The discrepancy could be due to project implementation differences, such as the fact because Ethiopia's northern regions are the most unstable during the time of instability transition, which could disrupt the implementation of mother and child health initiatives [37]. Furthermore, during the survey periods, drought and starvation in the northern part of the country may have contributed to the poor breast feeding in northern Ethiopia.

The local GWR analysis revealed that, being orthodox follower, poor wealth index, caesarian section, baby postnatal checkup, and small birth weight were spatially significant factors for delayed breastfeeding initiation in Ethiopia. In this study, poor household wealth status was found to be a geographically statistically significant predictive variable for breastfeeding initiation. Poor wealth index status had coefficients ranging from -0.175 to 0.318, with negative and positive strong relationships in different geographic locations. In Tigray, Amhara Afar, Southern Somali, Oromia, and Addis Ababa, it highly predicts the occurrence of late breastfeeding initiation. This could be attributable to a variety of factors, including differences in media access, lack of knowledge about the time to initiate breastfeeding, and the availability of health resources [38–41].

Similarly, caesarian section is a key spatial predictor of hotspots of delayed breastfeeding initiation across the region. Its strong and positive relationship with delayed initiation in

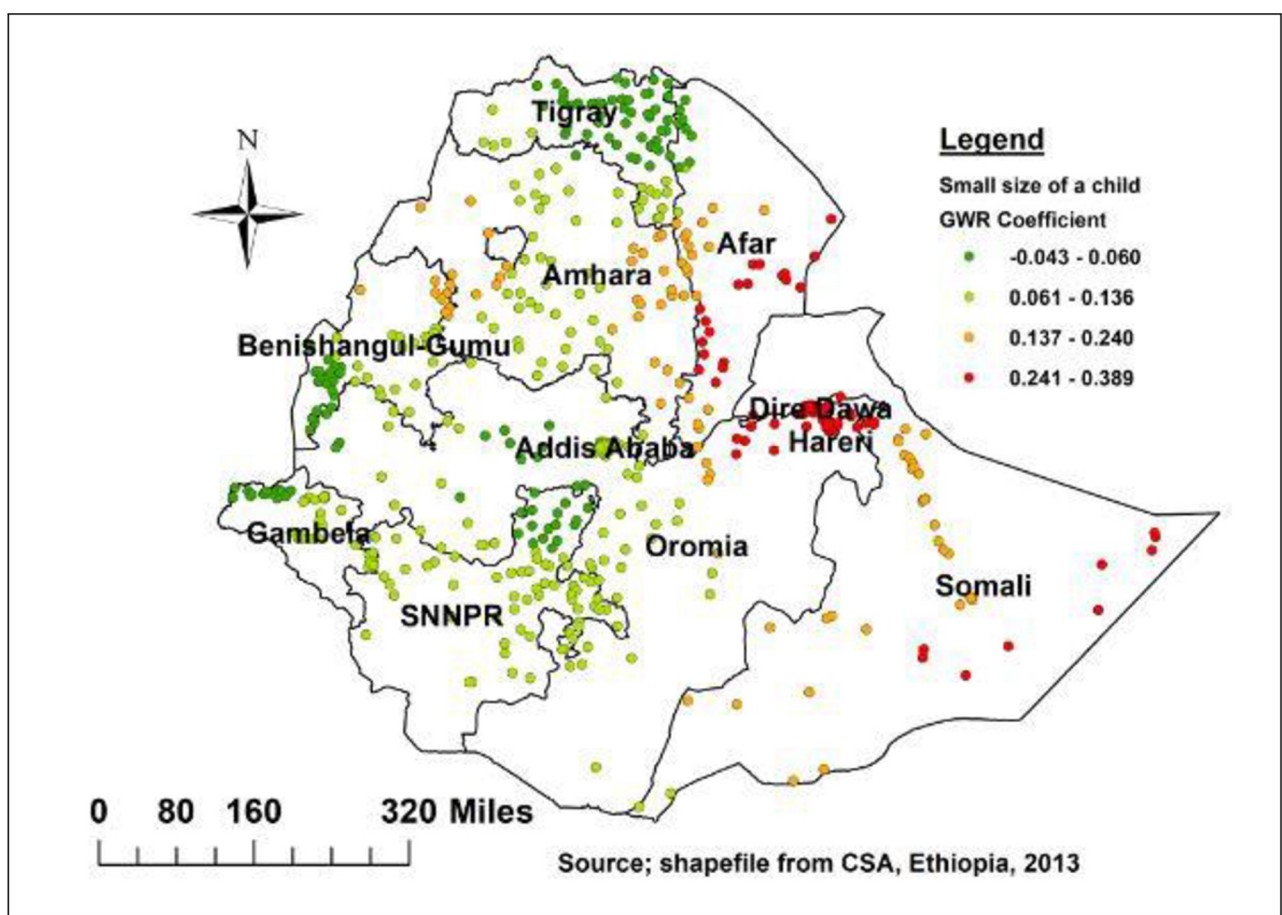

**Fig 10. Small size at birth geographic weighted regression coefficients to predict delayed breastfeeding in Ethiopia.**

Tigray, and border of Afar regions, whereas moderate and positive relationship in Oromia, SNNPR, Addis Ababa, Gambela, Amhara, central and southwest part of Afar. According to studies, the hospital practice of isolating infants from their mothers after caesarian section could explain and this could also be due to the mother's fatigue and pain following the birth [38, 42–44].

The baby receiving a postnatal check also strong and positive relationship was found in Oromia, SNNPR, Addis Ababa, Southeastern of Amhara, southwestern of Afar, and southwestern Somali region. As showed in Fig 9 children with small birth weight was strong predictor of delayed initiation of breastfeeding in eastern Somali region, southern and eastern parts of Afar. This finding could be explained by poor health personnel awareness that babies born with low birth weight should have skin-to-skin contact with their mothers to get early breastfeeding and avoid hypothermia [33, 45]. In addition, small sized neonate has worse suction ability breast seeking reflex and deglutition-respiration cycle [46, 47].

The strength of this study was using huge, nationally representative dataset, resulting in acceptable statistical power. Furthermore, the use of spatial and SaTScan based cluster analyses assisted in the detection of statistically significant high-risk clusters/hotspots of delayed breastfeeding initiation. A standard questionnaire was also used in the survey, which may have reduced the effect of measurement bias. As limitation, the location data values were relocated 1–2 km for urban areas and 5 km for rural areas this may have an impact on the precise

location of instances. We removed 34 clusters from the analysis because they lacked coordinated data, which may have influenced the estimated result. Besides, the cross-sectional nature of the study, we are unable to show the cause and effect relationship between dependent and independent variables and the survey replies may be disposed to a recall bias.

## Conclusion

In Ethiopia initiation of breastfeeding varies geographically across regions. A significant hotspot was identified in the Amhara, Afar, and Tigray regions. In GWR analysis orthodox religion, poor wealth index, caesarian section, baby postnatal checkup, and small size of a child at birth were spatially significant factors. Therefore, policymakers and health planners better to design an effective intervention program at hotspot regions and it is strongly essential that religious leaders educate women about early breastfeeding initiation.

## Acknowledgments

We would like to express our deepest thankfulness to Measure DHS, for providing the data for the study.

## Author Contributions

**Conceptualization:** Samuel Hailegebreal, Ermias Bekele Enyew.

**Data curation:** Samuel Hailegebreal, Binyam Tariku Seboka.

**Formal analysis:** Samuel Hailegebreal, Ermias Bekele Enyew.

**Methodology:** Yosef Haile, Binyam Tariku Seboka.

**Project administration:** Samuel Hailegebreal, Zeleke Abebaw Mekonnen.

**Software:** Zeleke Abebaw Mekonnen.

**Supervision:** Binyam Tariku Seboka, Tamiru Shibiru.

**Writing – review & editing:** Tamiru Shibiru, Shegaw Anagaw Mengiste.

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
