## [Decision Letter · Decision Letter 0]

10 Mar 2022

PONE-D-22-01403Modeling spatial determinates of initiation of breastfeeding in Ethiopia: A geographically weighted regression analysisPLOS ONE

Dear Dr.
Hailegebreal,

Thank you for submitting your manuscript to PLOS ONE. After careful consideration, we feel that it has merit but does not fully meet PLOS ONE’s publication criteria as it currently stands. Therefore, we invite you to submit a revised version of the manuscript that addresses the points raised during the review process. Please submit your revised manuscript by Apr 22 2022 11:59PM. If you will need more time than this to complete your revisions, please reply to this message or contact the journal office at plosone@plos.org. Please include the following items when submitting your revised manuscript:A rebuttal letter that responds to each point raised by the academic editor and reviewer(s). You should upload this letter as a separate file labeled 'Response to Reviewers'.A marked-up copy of your manuscript that highlights changes made to the original version. You should upload this as a separate file labeled 'Revised Manuscript with Track Changes'.An unmarked version of your revised paper without tracked changes. You should upload this as a separate file labeled 'Manuscript'.

We look forward to receiving your revised manuscript.

Kind regards,

Marcos Pereira

Academic Editor

PLOS ONE

Journal Requirements:

4. We note that Figures 3, 4, 5, 6, 7, 8 and 9 in your submission contain copyrighted images. All PLOS content is published under the Creative Commons Attribution License (CC BY 4.0), which means that the manuscript, images, and Supporting Information files will be freely available online, and any third party is permitted to access, download, copy, distribute, and use these materials in any way, even commercially, with proper attribution. For more information, see our copyright guidelines: http://journals.plos.org/plosone/s/licenses-and-copyright.

a. You may seek permission from the original copyright holder of Figures 3, 4, 5, 6, 7, 8 and 9 to publish the content specifically under the CC BY 4.0 license. 

Reviewers' comments:

Reviewer's Responses to Questions

**Comments to the Author**

1. Is the manuscript technically sound, and do the data support the conclusions?

Reviewer #1: Yes

Reviewer #2: Yes

2. Has the statistical analysis been performed appropriately and rigorously? 

Reviewer #1: Yes

Reviewer #2: Yes

3. Have the authors made all data underlying the findings in their manuscript fully available?

Reviewer #1: Yes

Reviewer #2: Yes

4. Is the manuscript presented in an intelligible fashion and written in standard English?

Reviewer #1: Yes

Reviewer #2: Yes

5. Review Comments to the Author

Reviewer #1: Dear,

The article is very well described and with robust analyses. But it would be interesting to insert a map in the methods describing the study area, identifying the locations geographically.

In figure 1, include the name of the "Y" axis with the unit of measurement (%).

Answer the question. What was the radius used to identify the Hot Spots? Include in methods.

Reviewer #2: This an interesting study on the spatial distribution and determinants of breastfeeding initiation in Ethiopia, using nationwide data from the 2016 Demographic and Health Survey. Although the manuscript presents robust spatial analyses and potential for publication, some points must be addressed and clarified:

The authors must present a more suitable Data Availability statement. Since the study used publicly available data from DHS, this statement may apply: “The data underlying the results presented in the study are available from (include the name of the third party and contact information or URL)”.

The manuscript is generally well written, but some English revision is needed to increase readability and correct typographical or grammatical errors.

Line 1: in the title and short title, please correct the typo “determinates” replacing it by “determinants”.

Line 34: In the abstract’s methods, please describe Global Moran's I.

Line 36-38: If possible, please present the GWR coefficient variation and respective confidential intervals or p-values for each of the risk factors associated with delayed breastfeeding initiation.

Please define the abbreviations when they first appear in the text: GRW (line 40), UNICEF (line 45), and ANC (line 60).

Lines 48-50: The two sentences could be better connected. Consider saying instead: “Early initiation of breastfeeding (EIBF) is an important pathway for reducing malnutrition and preventing mortality for young children (4,5) and reducing the risk of postpartum hemorrhage for the mothers (6)”.

Lines 51-52: Same content was said in the first sentence of the introduction. Please remove this sentence.

Lines 74-75: The authors mention nine region states and two municipalities where EDHS data were collected from. Were EDHS data representative of these regions and municipalities?

The authors must describe the study population and any inclusion or exclusion criteria applied in its selection. For example, children’s age range (0 to 23 months). This information must be described along with the study design section, which could be renamed “study design, setting and population”.

Line 81-83: Better description of the study variables is needed. Please provide more details on the collection of data on breastfeeding after birth. Also, explain why/how the independent variables were chosen, and how they were analyzed (categorical/numerical). If categorical, please specify each variable’s categories between parentheses. What do you mean by post-natal checkup? Please describe the variables considered in the household wealth index.

Line 93-95: What kind of spatial data were used in this analysis? Individual-level geocode data?

Line 98: Please correct: “Moran’s I value close to -1 indicates…”

How did the authors manage missing data? How were selected the final OLS and GWR models for the determinants of spatial variation in delayed breastfeeding initiation? AIC? Please clarify these points in the statistical analysis section.

Line 133: Specify the prevalence between parentheses.

Table 1: Please review the table’s title. Child sex is missing. Make clearer which variables are related to the mother and which ones are related to the child.

Table 2: What are the 611 observations? Clusters? If so, how did you arrived at this number of clusters? Make sure to clarify that in the results section when describing this OLR results. In addition, only important statistics must be kept in the table which in turn must be clear and self-explanatory.

Table 3: Define ACCs.

Line 247-250: As limitations, the authors must consider including the cross-sectional nature of the study; risk of recall bias regarding the breastfeeding initiation after birth, for example, almost half (46%) of the children were aged 12-23 months at the moment of the study.

In general, the discussion is very simplistic relying most in the description of the results. A major revision of this section must consider the contextualization of the results with the topic literature and the characteristics of the regions. A short description of the main regions, including demographic, socioeconomic, level of development (HDI), and prenatal and birth healthcare information, may help the authors to discuss the findings and clarify them to the broader readership.

6. PLOS authors have the option to publish the peer review history of their article (what does this mean?). If published, this will include your full peer review and any attached files.

Reviewer #1: No

Reviewer #2: No

---

## [Author Response · Author response to Decision Letter 0]

20 Apr 2022

To: PLOS ONE 

From: Samuel Hailegebreal 

Subject: A letter Accompanying Revision in Response to Editors and Reviewer Comments

Dear Editors 

The authors would like to thank the editorial team and team of reviewers for constructive and valuable comments. The authors are very happy to submit the revised version of the manuscript entitled “Modeling spatial determinants of initiation of breastfeeding in Ethiopia: A geographically weighted regression analysis” for its publication in your Journal. The comments of the editors and the reviewers were highly insightful and enabled us to greatly improve the quality of our manuscript. In this revised manuscript we made substantial changes to address your concerns in a point-by-point response. We are very keen to incorporate further comments, if any, for the betterment of the final manuscript.

Point by Point Response to - Editor Comments 

We note that Figures 3, 4, 5, 6, 7, 8 and 9 in your submission contain copyrighted images. All PLOS content is published under the Creative Commons Attribution License (CC BY 4.0), which means that the manuscript, images, and Supporting Information files will be freely available online, and any third party is permitted to access, download, copy, distribute, and use these materials in any way, even commercially, with proper attribution. For more information, see our copyright guidelines: http://journals.plos.org/plosone/s/licenses-and-copyright.

Authors’ response: Thank you editor for the concern. The figure in our manuscript including Figure.1 in the revised manuscript is not copyrighted rather we have done using ArcGIS and SaTScan software based on the shapefile of Ethiopia received from Ethiopian Central Statistical Agency (CSA) by explaining the purpose of the study and GPS data (longitude and latitude) from measure DHS program by explaining the objective of the study through online requesting and allow us to access the shapefile and GPS data. Now we cite the source of the shapefile since it is needed to explore the spatial distribution of home delivery. Therefore, the maps presented in our study are not copyrighted rather it was our spatial analysis result.

Point by Point Response to Reviewers 

Reviewer #1

1. The article is very well described and with robust analyses. But it would be interesting to insert a map in the methods describing the study area, identifying the locations geographically.

Authors’ response: We thank the reviewer for your great effort for the betterment of our work. We accepted your comments and modified accordingly.

 Figure.1 Map of study area, Ethiopia

2. In figure 1, include the name of the "Y" axis with the unit of measurement (%).

Authors’ response: We thank the reviewer for your great effort for the betterment of our work. We accepted your comments and modified accordingly.

Figure .2 prevalence of delayed initiation of breastfeeding across regions in Ethiopia, 2016.

3. Answer the question. What was the radius used to identify the Hot Spots? Include in methods

Authors’ response: Incremental spatial autocorrelation is a series of line graphs and their corresponding z-scores. The Z-score reflected the strength of clustering, and the peak was statistically significant which indicated where the spatial processes were promoted sound clustering. This tool can help you select an appropriate Distance Threshold or Radius for tools that have these parameters, such as hot spot analysis. Totally 10 distance bands were detected by a beginning distance of 121751 m, and first maximum peak (clustering) was observed at 196765.52 as shown below.

Reviewer #2: 

1. This an interesting study on the spatial distribution and determinants of breastfeeding initiation in Ethiopia, using nationwide data from the 2016 Demographic and Health Survey. Although the manuscript presents robust spatial analyses and potential for publication, some points must be addressed and clarified:

Authors’ response: We thank the reviewer for your great effort for the betterment of our work. We accepted your comments and modified accordingly.

2. The authors must present a more suitable Data Availability statement. Since the study used publicly available data from DHS, this statement may apply: “The data underlying the results presented in the study are available from (include the name of the third party and contact information or URL)”

Authors’ response: Availability of data and materials: The datasets analyzed during the current study are available from the DHS data set is available online and anyone can access it from (https://dhsprogram.com/data/available-datasets)

3. The manuscript is generally well written, but some English revision is needed to increase readability and correct typographical or grammatical errors.

Authors’ response: thank you for your comment we edited type/grammatical error 

4. Line 1: in the title and short title, please correct the typo “determinates” replacing it by “determinants”.

Authors’ response: We thank the reviewer for your comment “Modeling spatial determinants of initiation of breastfeeding in Ethiopia: A geographically weighted regression analysis”

5. In the abstract’s methods, please describe Global Moran's I.

Authors’ response: We thank the reviewer for your comment “Global Moran’s I statistic was used to measure whether delayed breastfeeding initiation was dispersed, clustered, or randomly distributed in study area.

6. If possible, please present the GWR coefficient variation and respective confidential intervals or p-values for each of the risk factors associated with delayed breastfeeding initiation.

Authors’ response: we thanks reviewer it’s possible to write the coefficient but we hope GWR map better to express the coefficient b/c most significant predictor had low coefficient(strong negative) and strong positive if we write the coefficient may mislead readers. The assumption of significance also found in our table. 

7. Please define the abbreviations when they first appear in the text: GRW (line 40), UNICEF (line 45), and ANC (line 60).

Authors’ response: We thank the reviewer for your great effort for the betterment of our work. We accepted your comments and modified accordingly (see revised manuscript)

8. Lines 48-50: The two sentences could be better connected. Consider saying instead: “Early initiation of breastfeeding (EIBF) is an important pathway for reducing malnutrition and preventing mortality for young children (4,5) and reducing the risk of postpartum hemorrhage for the mothers (6)”.

Authors’ response: We thank the reviewer for your great effort for the betterment of our work. We accepted your comments and modified accordingly (see revised manuscript)

9. Lines 51-52: Same content was said in the first sentence of the introduction. Please remove this sentence.

Authors’ response: We thank the reviewer for your comment. We accepted your comments and modified accordingly (see revised manuscript)

10. Lines 74-75: The authors mention nine region states and two municipalities where EDHS data were collected from. Were EDHS data representative of these regions and municipalities?

Authors’ response: We thank the reviewer for your comment. Ethiopia is located (3o-14oN, 33o – 48°E) in the Horn of Africa. It has 9 Regional states (Afar, Amhara, Benishangul-Gumuz, Gambela, Harari, Oromia, Somali, and Southern Nations, Nationalities, and People Region (SNNPR), and Tigray regions) and two cities (Addis Ababa and Dire-Dawa) administrations every five years (Fig.1). We used the 2016 EDHS data for this study which was conducted from January 18 2016 to June 27, 2016. In EDHS 2016, a two-stage stratified cluster sampling technique was employed using the 2007 Population and Housing Census (PHC) as a sampling frame. In the first stage, 645 EAs (202 in the urban area) were selected, and in the second stage, on average 28 households were systematically selected. We got the data from the EDHS dataset, which is only available (www.dhsprogram.com) through site requests(see revised manuscript). 

11. The authors must describe the study population and any inclusion or exclusion criteria applied in its selection. For example, children’s age range (0 to 23 months). This information must be described along with the study design section, which could be renamed “study design, setting and population”.

Authors’ response: we thank the reviewer comment. For this study, kids data set was used with a total weighted sample of 4169 women who ever breastfeed and who had living children less than 2 years of age (see revised manuscript) 

12. Line 81-83: Better description of the study variables is needed. Please provide more details on the collection of data on breastfeeding after birth. Also, explain why/how the independent variables were chosen, and how they were analyzed (categorical/numerical). If categorical, please specify each variable’s categories between parentheses. What do you mean by post-natal checkup? Please describe the variables considered in the household wealth index.

Authors’ response: we thanks the reviewer for insight comment revised accordingly

Study variables 

The outcome variable was delayed initiation of breastfeeding. It is put to the breast within the first hour of birth. It was measured based self-report of the mother and classified as "early" if it began within one hour, and "late /delayed" if it began later than one hour.

Independent variables

The variables were selected based on previous literatures review (22–24,24–26). In this analysis variables were recoded as follow mother’s age (“15–24 years”, “25–34 years”, “35–49 years”), marital status (“married”, “unmarried”), parity (“Primiparous”, “multiparous”, “grand multiparous”), place of residence (“rural” or “urban”), educational status(“no education”, “primary”, “secondary or above”), working status (“not working” or “working”), religion (“orthodox”, “Muslim”, “protestant”, “others” ), household wealth (“poor”, “middle”, “richer”), child age (“0–5 months”, “6–11 months”, “12–23 months”), baby post-natal check (“yes”/ “no”), antenatal care (“yes”, “no”), place of birth (“home” or “health facility”), mode of delivery (“caesarean section” or “vaginal”), birth weight (“small”, “average” “large”), birth order (“1-3”,”4-6” and “above 6”), media exposure (“yes”/ “no”) and child sex (“male” “female”). 

13. Line 93-95: What kind of spatial data were used in this analysis? Individual-level geocode data?

Authors’ response: The DHS program randomly displaced the GPS latitude/longitude positions (up to 2kms for urban and up to 5kms for rural clusters) for all DHS. Consequently, this study does not show the exact location of delayed breast feeding initiation in the study area.

14. Line 98: Please correct: “Moran’s I value close to -1 indicates…”

Authors’ response: corrected “Moran’s, I value close to -1 indicates that delayed breastfeeding initiation is dispersed, whereas Moran's I close to +1 indicates that delayed breastfeeding initiation is clustered, and Moran's I close to 0 indicates that delayed breastfeeding initiation is randomly distributed.”

15. How did the authors manage missing data? How were selected the final OLS and GWR models for the determinants of spatial variation in delayed breastfeeding initiation? AIC? Please clarify these points in the statistical analysis section.

Authors’ response: Features that contain missing values in the dependent or explanatory variables will be excluded from the analysis; however, you can use the Fill Missing Values tool to complete the dataset before running GWR. Replaces missing (null) values with estimated values based on spatial neighbors, space-time neighbors, or time-series values. But in this case no missing value observed.

Model selection- The Koenker Bp technique was used to see if the model could be used to do a spatially weighted regression analysis. When the Koenker statistics are significant (p-value<0.05), the GWR analysis is examined, which suggests the relationships between the dependent and independent variables change from place to place. The coefficients of explanatory variables in a correctly constructed OLS model should be statistically significant and have either a positive or negative sign. Multicollinearity (VIF<10) was also tested to rule out redundancy among independent variables.

The GWR equation is calibrated using data from nearby features, whereas the OLS equation uses data from all features. Finally, the corrected Akaike Information Criteria (AICc) and adjusted R-squared were used to compare models. The model with the lowest AICc value and the highest adjusted R-squared value was determined to be the best fit for the data.

16. Line 133: Specify the prevalence between parentheses.

Authors’ response: In the current study, the overall prevalence of delayed breastfeeding was 24.22 % [95 % CI: 22.94%, 25.55%]. The highest percentage of delayed breastfeeding was 56% [95% CI: 46%, 66%] seen in the Afar region (Fig.2).

17. Table 1: Please review the table’s title. Child sex is missing. Make clearer which variables are related to the mother and which ones are related to the child.

Authors’ response: we modified in the revised manuscript and also include child sex in the table (see the revised manuscript).

18. Table 2: What are the 611 observations? Clusters? If so, how did you arrived at this number of clusters? Make sure to clarify that in the results section when describing this OLR results. In addition, only important statistics must be kept in the table which in turn must be clear and self-explanatory.

Authors’ response: We appreciate the reviewer's insightful feedback. Clusters were marked by 611. Only 611 of the 645 EDHS for this analysis have spatial coordination (latitude/longitude). Here is found in limitation section in revised manuscript “As limitation, the location data values were relocated 1–2 km for urban areas and 5 km for rural areas this may have an impact on the precise location of instances. We removed 34 clusters from the analysis because they lacked coordinated data, which may have influenced the estimated result”

19. Table 3: Define ACCs.

Authors’ response: Akaike’s Information Criterion (AICc)

20. Line 247-250: As limitations, the authors must consider including the cross-sectional nature of the study; risk of recall bias regarding the breastfeeding initiation after birth, for example, almost half (46%) of the children were aged 12-23 months at the moment of the study.

Authors’ response: The study's strength is that it employed data from a large, nationally representative dataset, resulting in acceptable statistical power. Furthermore, the use of GIS and Sat Scan statistical analyses assisted in the detection of statistically significant high-risk clusters/hotspots of delayed breastfeeding initiation. A standard questionnaire was also used in the survey, which may have reduced the effect of measurement bias. As limitation, the location data values were relocated 1–2 km for urban areas and 5 km for rural areas this may have an impact on the precise location of instances. We removed 34 clusters from the analysis because they lacked coordinated data, which may have influenced the estimated result. Besides, the cross-sectional nature of the study, we are unable to show the cause and effect relationship between dependent and independent variables and the survey replies may be disposed to a recall bias.

21. In general, the discussion is very simplistic relying most in the description of the results. A major revision of this section must consider the contextualization of the results with the topic literature and the characteristics of the regions. A short description of the main regions, including demographic, socioeconomic, level of development (HDI), and prenatal and birth healthcare information, may help the authors to discuss the findings and clarify them to the broader readership.

Authors’ response: The author thanks reviewer comment. We modified our discussion parts (see the revised manuscript)

---

## [Decision Letter · Decision Letter 1]

14 Jul 2022

PONE-D-22-01403R1Modeling spatial determinants of initiation of breastfeeding in Ethiopia: A geographically weighted regression analysisPLOS ONE

Dear Dr. Hailegebreal,

Thank you for submitting your manuscript to PLOS ONE. After careful consideration, we feel that it has merit but does not fully meet PLOS ONE’s publication criteria as it currently stands. Therefore, we invite you to submit a revised version of the manuscript that addresses the points raised during the review process.

Your manuscript has been reassessed by the two reviewers from the previous round, whose reports can be found below. As you will see from the comments, the reviewers acknowledge that the manuscript has improved significantly, but there remain a small number of concerns which should be addressed before your manuscript is suitable for publication.

We look forward to receiving your revised manuscript.

Kind regards,

Joseph Donlan

Editorial Office

PLOS ONE

Journal Requirements:

Reviewers' comments:

Reviewer's Responses to Questions

**Comments to the Author**

1. If the authors have adequately addressed your comments raised in a previous round of review and you feel that this manuscript is now acceptable for publication, you may indicate that here to bypass the “Comments to the Author” section, enter your conflict of interest statement in the “Confidential to Editor” section, and submit your "Accept" recommendation.

Reviewer #1: All comments have been addressed

Reviewer #2: All comments have been addressed

2. Is the manuscript technically sound, and do the data support the conclusions?

Reviewer #1: Yes

Reviewer #2: Yes

3. Has the statistical analysis been performed appropriately and rigorously? 

Reviewer #1: Yes

Reviewer #2: Yes

4. Have the authors made all data underlying the findings in their manuscript fully available?

Reviewer #1: Yes

Reviewer #2: Yes

5. Is the manuscript presented in an intelligible fashion and written in standard English?

Reviewer #1: Yes

Reviewer #2: Yes

6. Review Comments to the Author

Reviewer #1: (No Response)

Reviewer #2: Thank you for addressing my comments. The authors have made important improvements in the manuscript, making it much clearer and suitable for publication.

1. It is not clear if the authors used the “fill missing value” tool to complete the dataset with estimated values or if they just excluded the observations with missing value in any of the study variables. In the section “Data management and Statistical analysis”, please clarify how you handled missing data.

2. Do you really need all those statistics in table 2 in order to interpret the results? Please leave in the table only the essential statistics.

7. PLOS authors have the option to publish the peer review history of their article (what does this mean?). If published, this will include your full peer review and any attached files.

Reviewer #1: **Yes: **PhD Marcio Natividade (ISC/UFBA)

Reviewer #2: No

---

## [Author Response · Author response to Decision Letter 1]

15 Jul 2022

To: PLOS ONE 

From: Samuel Hailegebreal 

Subject: A letter Accompanying Revision in Response to Editors and Reviewer Comments

Dear Editors 

The authors would like to thank the editorial team and team of reviewers for constructive and valuable comments. The authors are very happy to submit the second revised version of the manuscript entitled “Modeling spatial determinants of initiation of breastfeeding in Ethiopia: A geographically weighted regression analysis” for its publication in your Journal. The comments of the editors and the reviewers were highly insightful and enabled us to greatly improve the quality of our manuscript. In this second revision manuscript we made substantial changes to address your concerns in a point-by-point response. We are very keen to incorporate further comments, if any, for the betterment of the final manuscript.

Point by Point Response to - Editor Comments 

Point by Point Response to Reviewers 

Author response: thank you for your comment. We author updated reverence no 1, 2, 3, 27, 28 in the revised manuscript 

Reviewer #1

1. It is not clear if the authors used the “fill missing value” tool to complete the dataset with estimated values or if they just excluded the observations with missing value in any of the study variables. In the section “Data management and Statistical analysis”, please clarify how you handled missing data.

Authors’ response: Thank you for your input. A variable with a "missing value" should have a response but does not because the question was not asked (due to interviewer error) or the responder did not want to answer. We use the STATA drop command in conjunction with a logical / conditional expression to remove missing values from our analysis.

2. Do you really need all those statistics in table 2 in order to interpret the results? Please leave in the table only the essential statistics.

Authors’ response: Thank you for the comment we accept and corrected in the revised version 

Variable Coefficient Robust standard error Robust t statistics Robust probability VIF

Intercept 0.06 0.020 3.09 < 0.01 ------

Orthodox 0.12 0.024 5.05 < 0.01 1.13

Proportion poor 0.17 0.028 6.09 < 0.01 1.27

Proportion of cesarean delivery 0.22 0.079 2.83 < 0.01 1.12

Proportion of baby postnatal checkup 0.13 0.063 2.08 < 0.05 1.09

Proportion of small size at birth 0.20 0.046 4.24 < 0.01 1.08

Ordinary least square regression diagnostics 

Number of Observations: 611 Adjusted R-Squared 0.137

Joint F-Statistic 20.37 Prob(>F), (5,605) degrees of freedom < 0.01

Joint Wald Statistic: 100.27 Prob(>chi-squared), (5) degrees of freedom < 0.01

Koenker (BP) Statistic 22.100 Prob(>chi-squared), (5) degrees of freedom < 0.01

Jarque-Bera Statistic 39.838 Prob(>chi-squared), (2) degrees of freedom < 0.01

---

## [Decision Letter · Decision Letter 2]

11 Aug 2022

PONE-D-22-01403R2Modeling spatial determinants of initiation of breastfeeding in Ethiopia: A geographically weighted regression analysisPLOS ONE

Dear Dr. Hailegebreal,

Thank you for submitting your manuscript to PLOS ONE. After careful consideration, we feel that it has merit but does not fully meet PLOS ONE’s publication criteria as it currently stands. Therefore, we invite you to submit a revised version of the manuscript that addresses the points raised during the review process. In response to reviewer #2's comment on missing values (see below), you have provided an explanation in the response letter but you have not included this explanation in the "Date management and statistical analysis" section of the Methods. Please could you ensure that you update this section accordingly and then resubmit. --Reviewer #2 comments1. It is not clear if the authors used the “fill missing value” tool to complete the dataset with estimated values or if they just excluded the observations with missing value in any of the study variables. In the section “Data management and Statistical analysis”, please clarify how you handled missing data.

--

We look forward to receiving your revised manuscript.

Kind regards,

James Mockridge

Staff Editor

PLOS ONE

Journal Requirements:

Reviewers' comments:

Reviewer's Responses to Questions

**Comments to the Author**

1. If the authors have adequately addressed your comments raised in a previous round of review and you feel that this manuscript is now acceptable for publication, you may indicate that here to bypass the “Comments to the Author” section, enter your conflict of interest statement in the “Confidential to Editor” section, and submit your "Accept" recommendation.

Reviewer #1: All comments have been addressed

2. Is the manuscript technically sound, and do the data support the conclusions?

Reviewer #1: Yes

3. Has the statistical analysis been performed appropriately and rigorously? 

Reviewer #1: Yes

4. Have the authors made all data underlying the findings in their manuscript fully available?

Reviewer #1: Yes

5. Is the manuscript presented in an intelligible fashion and written in standard English?

Reviewer #1: Yes

6. Review Comments to the Author

Reviewer #1: Congratulations to the actors. I believe that the manuscript will contribute in a relevant way to studies in the field of breastfeeding in the reflection of social determination.

7. PLOS authors have the option to publish the peer review history of their article (what does this mean?). If published, this will include your full peer review and any attached files.

Reviewer #1: No

---

## [Author Response · Author response to Decision Letter 2]

11 Aug 2022

1. In response to reviewer #2's comment on missing values (see below), you have provided an explanation in the response letter but you have not included this explanation in the "Date management and statistical analysis" section of the Methods. Please could you ensure that you update this section accordingly and then resubmit?

Author response: When a variable has a "missing value," it should have a response but does not, either because the interviewer accidentally left out the question or because the respondent declined to answer. We remove missing values from our analysis by using the STATA drop command in combination with a logical or conditional statement.

---

## [Editor Report · Decision Letter 3]

16 Aug 2022

Modeling spatial determinants of initiation of breastfeeding in Ethiopia: A geographically weighted regression analysis

PONE-D-22-01403R3

Dear Dr. Hailegebreal,

We’re pleased to inform you that your manuscript has been judged scientifically suitable for publication and will be formally accepted for publication once it meets all outstanding technical requirements.

Kind regards,

James Mockridge

Staff Editor

PLOS ONE

---

## [Editor Report · Acceptance letter]

2 Sep 2022

PONE-D-22-01403R3 

Modeling spatial determinants of initiation of breastfeeding in Ethiopia: A geographically weighted regression analysis 

Dear Dr. Hailegebreal:

I'm pleased to inform you that your manuscript has been deemed suitable for publication in PLOS ONE. Congratulations! Your manuscript is now with our production department. 

Kind regards, 

on behalf of

Dr James Mockridge 

Staff Editor

PLOS ONE